# A Novel High-Voltage Gain Step-Up DC–DC Converter with Maximum Power Point Tracker for Solar Photovoltaic Systems

Rashid Ahmed Khan [1], Hwa-Dong Liu [2], Chang-Hua Lin [1,*], Shiue-Der Lu [3], Shih-Jen Yang [4] and Adil Sarwar [5]

[1] Department of Electrical Engineering, National Taiwan University of Science and Technology, Taipei 106, Taiwan
[2] Undergraduate Program of Vehicle and Energy Engineering, National Taiwan Normal University, Taipei 106, Taiwan
[3] Department of Electrical Engineering, National Chin-Yi University of Technology, Taichung 411, Taiwan
[4] College of Electrical Engineering and Computer Science, National Taipei University, New Taipei City 237, Taiwan
[5] Department of Electrical Engineering, ZHCET, Aligarh Muslim University, Aligarh 202002, India
[*] Correspondence: link@mail.ntust.edu.tw; Tel.: +886-2-2730-3289

**Abstract:** In order to generate electricity from solar PV modules, this study proposed a novel high-voltage gain step-up (HVGSU) DC–DC converter for solar photovoltaic system operation with a maximum power point (MPP) tracker. The PV array can supply power to the load via a DC–DC converter, increasing the output voltage. Due to the stochastic nature of solar energy, PV arrays must use the MPPT control approach to function at the MPP. This study suggests a novel HVGSU converter that uses the primary boost conversion cell and combines switched capacitors and voltage multiplier cells. The proposed topology is upgradeable for high-voltage gain step-up and can be incorporated as well. A clamp circuit reuses the energy that leaks out so that the switch voltage stress and power loss are kept to a minimum. One thing that makes it stand out is that the voltage stress on the diodes and switch stays low and constant even as the duty cycle changes. Additionally, the inductor greatly reduces the diodes' reverse recovery losses. There is a lot of information about steady-state analyses, operation principles, and design guidelines. A prototype circuit is built to test the maximum power point tracking operation with voltage conversion from 20–40 V to 380 V at 150 W. The results of the experiments support the theoretical analysis and claimed benefits. The proposed converter has the ability to track the maximum power point and a high conversion efficiency over a wide range of power. A weighted efficiency of 90–96% is shown by the prototype.

**Keywords:** high-voltage gain step-up DC–DC converter; hill climbing algorithm; maximum power point tracker; non-isolated topology

## 1. Introduction

The widespread use of conventional fossil fuels has reduced fuel reserves and impacted the atmosphere, creating pollution and the greenhouse effect. Since fossil fuels can't be replenished, we face the challenge of an energy crisis. It is essential to generate clean and renewable energy to replace traditional fossil fuels. Solar energy is promising due to its availability and the widespread application of photovoltaic generating as a method of consumption. As a result, numerous techniques have been used to enhance the effectiveness of the standalone PV system. Power generated by renewable energy sources (RES) has a low DC voltage level and needs to be increased using DC–DC converters. As a result, recent years have seen research into high step-up DC–DC converters [1–5]. To make RES more useful, such different types of converters should be able to change voltage with a very high gain while still meeting the other quality requirements [6–8]. Since the conventional boost converter (CBC) has a simple design, it seems to be the best choice

for boosting the voltage of RESs [9]. However, the parasitic effect of passive elements at high-duty cycles restricts CBC's ability to increase voltage adequately [10]. All of the traditional converters have this characteristic. As a result, research is conducted on converter combinations with boosting voltage approaches, which increase the voltage gain at low-duty cycles [11–15]. DC–DC converters commonly incorporate boosting circuits such as switching capacitors, multiplier cells, coupled inductors, and high-frequency transformers to reach greater voltage levels [16,17]. In paper [18], the two most well-known stepping-up methods cascading and interleaving are discussed, and both methods have their benefits and drawbacks to be consider. While the theory and implementation of the cascading approach are straightforward, the escalating number of components needed inevitably reduces the efficiency and restricts the power density of the converter. The fundamental advantage of interleaved converters for fuel cell (FC) applications is the decrease of the input current ripple. Since the interleaving technique is inadequate for the rapidly approaching ultra-high-voltage gain, the boosting cells must be used at various points in the process [18]. Aside from those above, switching capacitors and multiplier cells are two more widely utilized approaches in high step-up DC–DC converters [19,20]. Components associated with these techniques are typically situated on the output port of converters and integrated with other approaches. The major reason they are used in the high step-up converter is that they increase the voltage gain by a factor of two or three compared to converters with a small number of components. To improve the voltage gain of the boost converter, coupled inductors (CI) were added to the circuitry in work [18]. Since the effect on the voltage gain is good, these converters isolate certain topology components. This sort of converter uses fewer magnetic cores by winding many coils around a single core [21,22]. Some of the proposed converters use CIs to boost the voltage gain. However, this comes at the expense of a higher conduction loss and larger leakage inductance due to the high turn ratio required [19]. Therefore, they should be strategically placed in the proposed converter layouts to maximize the voltage gain despite the low turn ratio. One more benefit of using CI is that the voltage stress on components in some situations can be reduced. Choosing a suitable converter as the foundation of the setup is yet another method for increasing the voltage gain in a DC–DC converter. The quadratic boost converter (QBC) can improve the voltage gain similar to a quadratic function of the duty cycle when two CBCs are cascaded [12,15]. In the converter discussed in [13], the power switch is subjected to a high-voltage stress equal to the output voltage, which results in dispersing the voltage stress across two power switches and reduces the voltage stress on the switches. Due to their adaptability for novel combinations, continuous input current, the presence of ground between the source and load, and visible voltage gain for the foundation of new configurations, several studies have suggested improved architectures for quadratic boost converters. In addition to their use in hybrid inverters, quasi-Z-source and Z-source converters are the other two types of converters for high step-up applications [23–26]. The Z-source converter is restricted in its RES applications by its discontinuous input current, poor voltage gain, and shared ground between the load and source. Although a quasi-Z-source converter can have a common grounded configuration and receive a constant input current, the voltage gain is low, requiring numerous voltage boosting stages for functioning similar to an ultra-high-voltage gain converter. The research on high step-up DC–DC converters has shown some important things they all have in common. These are a lower number of components, high-voltage gain, common ground between source and load, high efficiency, low voltage stress on the circuit components, and a steady input current.

A schematic diagram of the microgrid system is displayed in Figure 1. High-gain DC–DC converters are used to convert the low voltage (20–40 V) produced by the battery, solar photovoltaic (PV), or fuel cell into a high DC voltage (350–400 V) and act as an intermediary between the load and the source. In a DC microgrid, a high-gain DC–DC converter keeps the DC link voltage steady at a predetermined level [27]. Due to their high

power density but low voltage rating, supercapacitors are often used in conjunction with a high-gain DC–DC converter in modern DC microgrids.

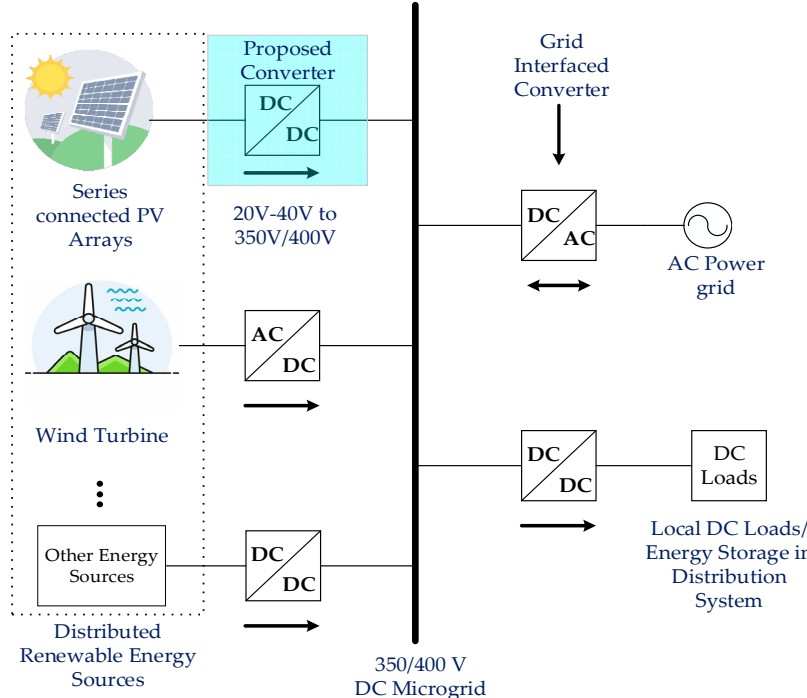

**Figure 1.** Schematic diagram of a microgrid system.

Level three rapid charging for electric vehicles (EVs) uses high-gain converters. In islanded mode functioning of a DC microgrid, AC loads can be fed by combining a high-gain converter with an inverter. High-gain converters have become more popular, because conventional boost converters and their variations are inefficient and have to work with a high-duty cycle to achieve a high gain [27]. Additionally, there is an issue with the diode reverse recovery when the duty cycle increases. One can essentially categorize these converters as either non-isolated or isolated. To prevent the current from flowing directly from the output to the input, isolated converters physically separate the circuit into two halves. Using a high-frequency transformer accomplishes this at the expense of the converter's size and cost [28–30]. In high-power applications and those requiring a common ground between the source and the load, isolated topologies are preferred. Where input and output isolation are not necessary, non-isolated converters are preferred. It is possible to divide them into coupled and uncoupled types [31]. While coupled inductor topologies have the potential to generate a very high voltage gain with minimal stress on the switch, issues with leakage inductance can cause huge voltage spikes across the switch, necessitating the design of a clamped circuit. Cascading two or more converters creates a high-gain capacitor diode voltage multiplier cell. Some novel and modified topologies include a voltage multiplier circuit (VMC) designed from switched inductors and capacitors to boost the gain of the converter. A high gain can be achieved with low-duty cycles and less stress on semiconductor switches using the cubic boost and quadratic boost converter (QBC) topologies [32–34]. Although the efficiency may drop with increasing the current at greater duty ratios. The inductor current ripple and switch stress were both reduced in a novel QBC suggested in [35]. Interleaved boost converters were also employed to achieve a high output voltage and higher efficiency [36] with the reduced number of switches. It is possible to attain a high gain with a multiphase interleaved converter by incorporating a Z-source network. Since the ripple in the input current is so low, there is no requirement for an input ripple filter. The issue with interleaved converter topologies is that they require a voltage boost circuit at the end to make the converter

more efficient [37]. In [38], several alternative converter topologies based on the well-known Cockcroft–Walton voltage multiplier cell are studied. Reference [39] introduced a novel high-gain single-switch single-ended primary inductor converter (SEPIC). In [40], we saw another novel SEPIC converter employing a combination of buck/boost topology. A single input to multiport outputs of varying voltages is a simple and efficient technique. Compared to single-input single-output (SISO) converters [41,42], these are superior in terms of voltage control capabilities and stress across the output capacitor. It is possible to use them independently as SISO converters. In [43], the voltage lift method (VLM) was applied to the problem of designing a quadratic boost converter. The VLM approach was used to achieve a specific gain [44]. However, the converter employed a pair of switches cycled through in a unique pattern. In [45], a high-gain hybrid converter comprised of voltage multiplier cells (VMCs) and switching capacitor cells was reported. It eliminated problems, such as excessive voltage and current stress on the power devices. The authors in [46] proposed a converter with an identical gain but with a different stress on the two switches. They used a switching inductor voltage multiplier and a diode voltage capacitor multiplier to accomplish this. Using a similar notion of a switched/series capacitor with six switches, Ref. [47] presented a high step-down interleaved converter. Reference [48] detailed a unique converter based on a switched impedance network. The gain was increased at lower duty ratios using a voltage double-switch capacitor circuit. The gain can go up significantly with a multistage structure, but efficiency can decrease because of more components. In [49], a hybrid zeta boost converter with a switched inductor was suggested but did not have a very high-voltage gain. In [50], two switches were used to make a new transformerless active switch network, but the gain was less than proposed in this paper. Paper [51] discussed a topology based on a switched capacitor circuit. Despite employing two switches and two inductors, the voltage gain discussed in [52], the switched inductor topology was small. The high-voltage gain reported in [53] resulted from an improved SEPIC converter. The voltage gain of the converter in [54] was improved by employing a parallel input series output (PISO) approach. Diodes and switches now have a lower stress voltage as a result. In [55], an active-clamp topology was introduced, and this topology used a voltage-doubling active-clamp circuit consisting of two switches, two diodes, and two capacitors. Although only one active-clamp circuit is required in the on/off states, this design provides all the benefits of a forward/flyback DC/DC converter. Non-isolated high step-up DC–DC converters with a built-in transformer topology were introduced in [56] as an improved substitute to interleaved step-up DC/DC converters. Even though a transformer is used, there is no electrical isolation. According to [57], a series-resonant voltage-doubling active-clamp circuit was used to create a step-up DC–DC converter with zero-voltage switching (ZVS) capability.

In [58], a single-switch quadratic boost converter with two-stage boosting capabilities was presented to help reduce the number of switches. Another soft-switching converter based on a quadratic boost converter with a single resonant network was proposed in [59]. According to [60], a comprehensive study was conducted on a converter with a large conversion range, with the gain factor changing depending on the duty ratio and the number of switching networks used. In [61], a high-gain boost converter with a maximum efficiency of 93% and little voltage stress on the components was presented for microgrid applications. In [62], a thorough design of a cascaded DC–DC converter with efficiency optimization was given. To achieve a gain of approximately 10 at a 0.7 duty ratio operation, a novel quadratic boost converter design was presented in [63], employing only one active switch. According to [64], a similar design was predicted to provide good gain while putting a minimal voltage load on solid-state devices. Further extension of the quadratic boost converter with a voltage multiplier circuit was analyzed in [65,66], which lowers the input current ripple and the voltage stress on devices, resulting in the improved overall performance of the converter.

To address the abovementioned issues, an HVGSU DC–DC converter is developed in this study. The proposed converter has a high voltage gain by selecting the appropriate

duty cycle for the power switches and designing appropriate inductor and capacitor values. It has the following benefits: (i) The power switch of the proposed HVGSU DC–DC converter operates with different duty ratios to assist in achieving a high-voltage gain; (ii) the stored inductor energy is supplied to the load without the use of an additional clamping circuit; (iii) the voltage gain achieved by the proposed converter is greater than that of the other boost converter topology, and the voltage gain vs. duty ratio plot of the recently proposed topology is shown in Figure 2; (iv) a reduced voltage stress on switches based on the percentage output voltage; (v) the proposed converter is capable of achieving a high-voltage gain at a very low-duty cycle; and (vi) the output of the proposed converter is constant, with applications such as harnessing renewable energy, charging applications, microgrids, DC distribution systems, and uninterruptible power supplies (as shown in Figure 1). The passive clamp can absorb the energy loss due to the inductor's leaking inductance to prevent an excessive voltage spike at the switch. The passive clamp and voltage multiplier circuit arrangement boost the voltage gain. The advantages of this converter include its high efficiency, low switching loss, high-voltage gain at a low-duty cycle, low turn ratio for the inductor, low-voltage stress on the switch, and low-voltage stress on the diodes.

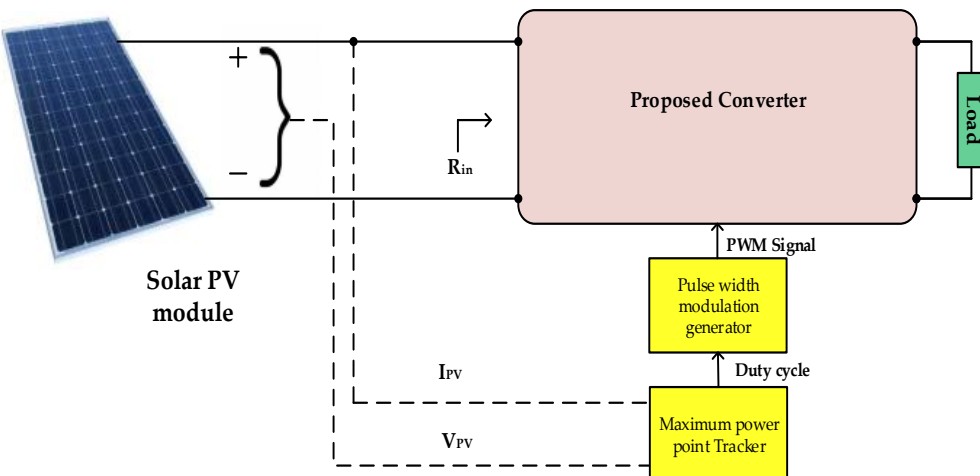

**Figure 2.** A simplified block schematic of the MPPT circuitry found in a solar PV module.

A high-voltage gain DC–DC converter based on diode–capacitor VMC cells is proposed in this paper. The proposed converter has a high-voltage gain ratio and a constant output voltage with a low ripple current and a single switch. Compared to conventional boost converters and other high-gain converters, the voltage stress across semiconductor devices is low, and the required energy storage is less. This paper is divided into eight parts, the first section of which serves as an introduction. Integration of the maximum power point tracker with the proposed HVGSU DC–DC converter is discussed in Section 2. Section 3 covers the proposed converter's structure and its operation modes. For the simulation results and detailed waveforms of the HVGSU DC–DC converter, and to verify the analysis and design, a 150 W prototype was implemented. The experimental results obtained using the prototype are presented in Section 4. Section 5 includes the conclusion and future works.

## 2. Integration of Maximum Power Point Tracker with the Proposed HVGSU DC–DC Converter

Typically, the output efficiency of solar PV modules ranges between 15 and 22%. Numerous factors, including the orientation of the PV module, topographical position, temperature, humidity, and rate of dust accumulation, contribute to a decrease in the output efficiency of the PV module. Power electronic devices accompanied by a maximum power point (MPP) tracker are used to augment or improve the efficiency of solar PV modules. The MPPT controllers extract the highest possible power from solar PV modules. The

use of MPPT improves the output performance of the solar PV module greatly. Various techniques are being built to effectively track the MPP. Existing MPPTs have the primary drawback or constraint of tracking the maximum power slowly. Consequently, the output efficiency of solar PV modules degrades. Figure 2 depicts a fundamental block diagram of an MPPT-integrated solar PV module.

Maximum power point tracking is an algorithm included in change controllers and is utilized to harvest the maximum amount of available power from PV modules under specific circumstances. Different algorithms are used to control the duty cycle converter [67–72]. Maximum power point refers to the voltage at which a PV module is capable of producing the most amount of power. The maximum power can change depending on the amount of solar radiation, the surrounding temperature, and the temperature of the solar cell.

The hill climbing (HC) algorithm is widely used in practical PV systems due to its ease of use; the fact that it does not require the study or modeling of source characteristics; and the fact that it is able to account for characteristic drifts caused by aging, shadowing, or other operating irregularities. The first step is to determine the voltage ($V_{pv}(n)$) and current ($I_{pv}(n)$) that are currently being produced by the PV array. Therefore, the generated power ($P_{pv}(n)$) can be calculated and compared to its value that was calculated in the previous iteration. Depending on the comparison result, the sign of a "slope" is either complimented or remains unaltered, and the PWM output duty cycle is modified appropriately. Figure 3 presents the flow chart of the HC algorithm. This algorithm is effective for environments with uniform sunshine but not for environments with shade. This research employs the novel high-voltage gain step-up (HVGSU) DC–DC converter, which has a high boost effect, to enable parallel operations of solar PV modules. Even if the shading issue arises, the efficiency will not be affected. The HVGSU converter proposed in this experiment in conjunction with the HC algorithm can operate in a partial shade environment and mitigate the algorithm's inadequacies.

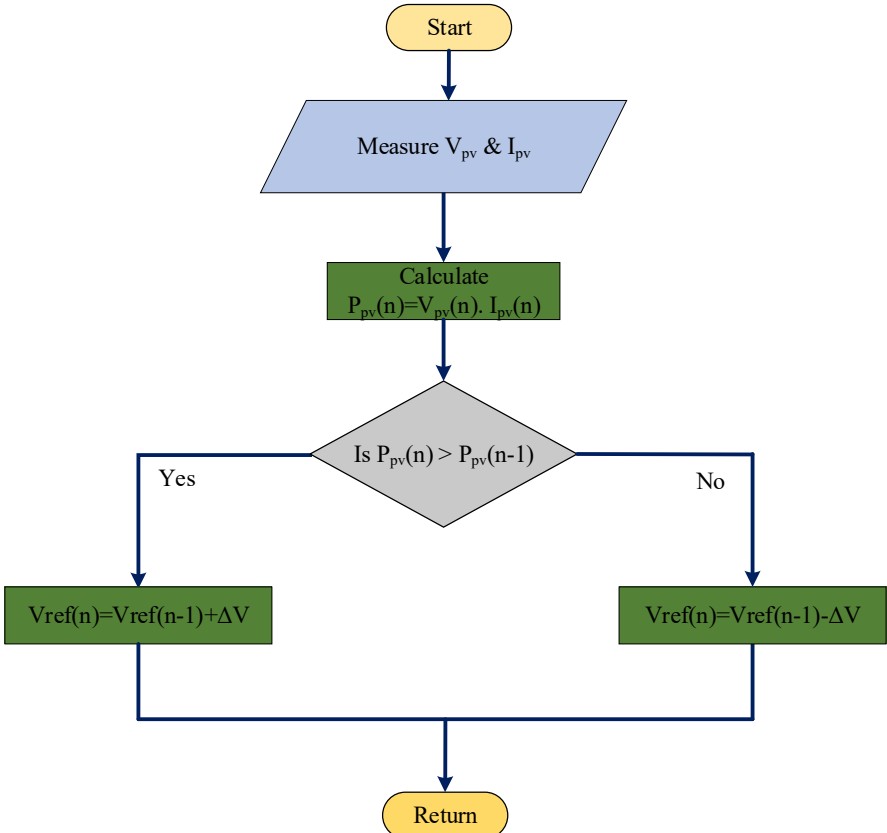

**Figure 3.** Flow chart of the MPPT algorithm based on the HC algorithm.

The flowchart of the constant voltage control approach paired with the HC MPPT control strategy is depicted in Figure 4. First, detect $V_{pv}$, $I_{pv}$, and $V_o$ and then use the constant voltage control method to make $V_o$ operate between 370 V and 390 V. Second, the constant voltage control approach is incapable of making the output voltage $V_o$ operate between 370 V and 390 V, so then implement the HC MPPT control strategy to acquire the maximum power. However, the HC MPPT control strategy uses the perturbation mechanism to perform the MPPT. Therefore, the output voltage $V_o$ is set between 360 V and 400 V. Lastly, if the HC MPPT control method is unable to control the output voltage $V_o$ between 360 V and 400 V, it will be unable to supply the voltage level required by the DC microgrid, and the solar photovoltaic module's output power will decrease. At this point, it will terminate operations, and the control technique will restart when it detects a $V_{pv}$ greater than 20 V.

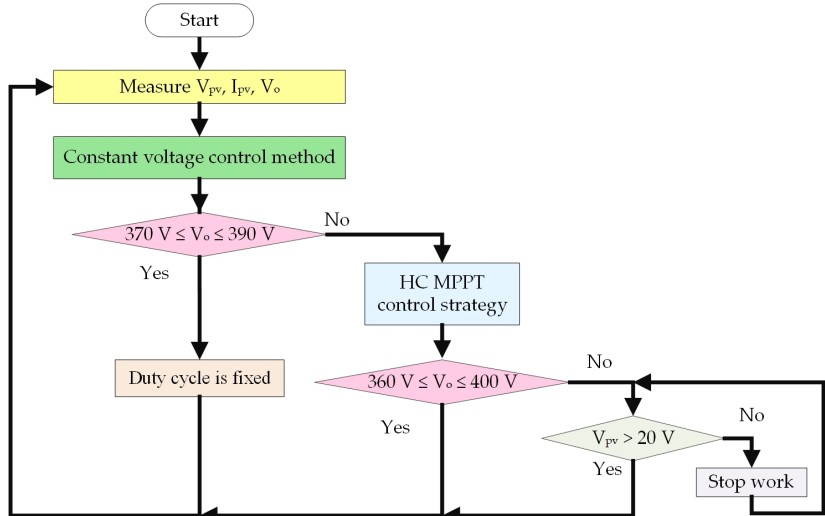

**Figure 4.** Flow chart of the constant voltage control method combined with the HC MPPT control strategy.

## 3. Structure of the Proposed Converter

The structure of the proposed novel high-voltage gain step-up (HVGSU) DC–DC converter with ultra-high-voltage gain. is shown in Figure 5, which consists of one switch; an inductor ($L_1$) in series with the input source; and two switched inductors ($L_2$ and $L_3$); diodes ($D_1$, $D_2$, $D_3$, $D_4$, $D_5$, $D_6$, and $D_7$); and capacitors ($C_1$, $C_2$, $C_3$, $C_4$, and $C_o$). Due to its location in the center of the converter, the switch is subject to minimal voltage stress. The switch separates the converter in half.

The generalized circuit diagram of the proposed topology is given in Figure 5. The circuit comprises three inductors, i.e., $L_1$, $L_2$, and $L_3$, and five capacitors, $C_1$, $C_2$, $C_3$, $C_4$, and Co, including seven diodes $D_1$, $D_2$, $D_3$, $D_4$, $D_5$, $D_6$, and $D_7$ and one power MOSFET switch. The voltage multiplier cell (VMC) consists of two inductors, $L_2$ and $L_3$, one capacitor $C_2$, and two diodes, $D_2$ and $D_3$. The integrated switch capacitor network comprises two capacitors, $C_3$ and $C_4$, and two diodes, $D_4$ and $D_5$.

Figure 6 displays the steady-state waveform of the proposed converter across all passive components and the current following through all active devices. During period DT, the switch (SW) is turned on; the inductor currents $I_{L1}$, $I_{L2}$, and $I_{L3}$ rise until $I_{L1max}$, $I_{L2max}$, and $I_{L3max}$, respectively; and the current across the switch, $I_{SW}$, also decreases. The capacitors $C_1$ and $C_2$ start charging, and the diodes $D_1$, $D_2$, $D_3$, and $D_4$ are forward-biased; hence, the diode currents $I_{D1}$, $I_{D2}$, $I_{D3}$, and $I_{D4}$ rise until the $I_{D1max}$, $I_{D2max}$, $I_{D3max}$, and $I_{D4max}$. The diodes $D_5$, $D_6$, and $D_7$ are reversed-biased during this period. Hence, there is no diode currents $I_{D5}$, $I_{D6}$, and $I_{D7}$. During period (1D) T, the switch is turned off as a result, and the inductor currents $I_{L1}$, $I_{L2}$, and $I_{L3}$ fall until the $I_{L1min}$, $I_{L2min}$, and $I_{L3min}$, respectively.

During the same time interval, the current across the switch SW is zero, and the capacitors $C_1$ and $C_2$ start discharging while the capacitors $C_3$ and $C_4$ start charging. The diodes $D_1$, $D_2$, $D_3$, and $D_4$ are reverse-biased; hence, the diode currents $I_{D1}$, $I_{D2}$, $I_{D3}$, and $I_{D4}$ fall until the $I_{D1min}$, $I_{D2min}$, $I_{D3min}$, and $I_{D4min}$. The diodes $D_5$, $D_6$, and $D_7$ are forward-biased during this period. Hence, there are no diode currents $I_{D1}$, $I_{D2}$, $I_{D3}$, and $I_{D4}$. The same cycle is repeated for the next period.

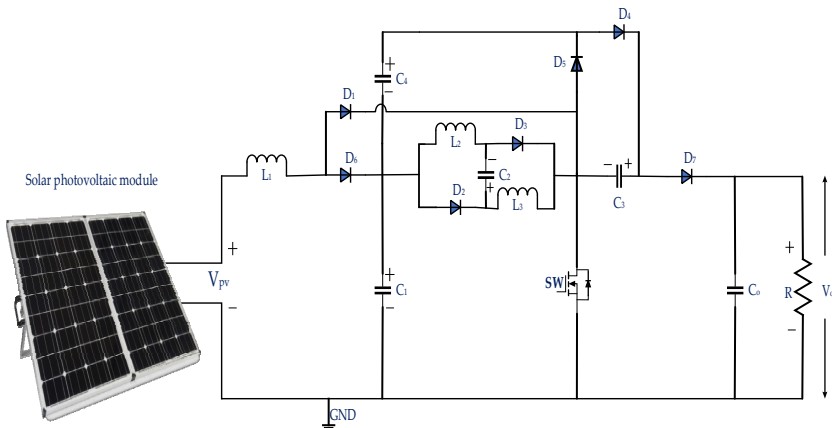

**Figure 5.** Proposed topology circuit with PV module configurations.

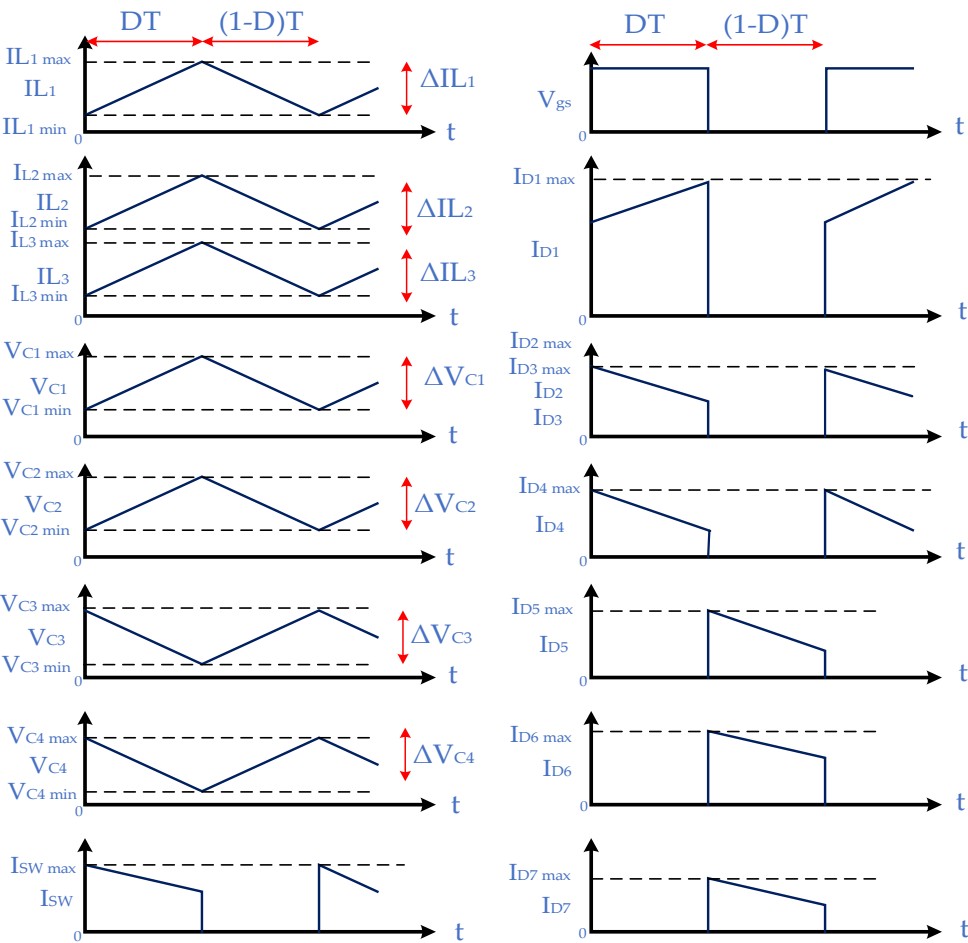

**Figure 6.** Steady-state waveform of the proposed HVGSU converter.

### 3.1. Operation of Proposed HVGSU Converter Configuration (Mode 1: SW Turn On)

During mode 1 of operation, as shown in Figure 7, the capacitor $C_1$ charges each part of the VMC. While $C_3$ is connected in a series with the VMC cell, it gets charged by the sum of the potentials of $C_4$ and $C_1$. On the other hand, the source voltage will be stored in the inductor $L_1$ through diode $D_1$. Diode $D_7$ will have a reverse bias, and the load current will come from the output capacitor. In this mode, inductors $L_1$, $L_2$, and $L_3$ are energy storage inductors, and diodes $D_1$, $D_2$, $D_3$, and $D_4$ are only forward-biased. During this mode of operation, all other diodes will be biased in reverse.

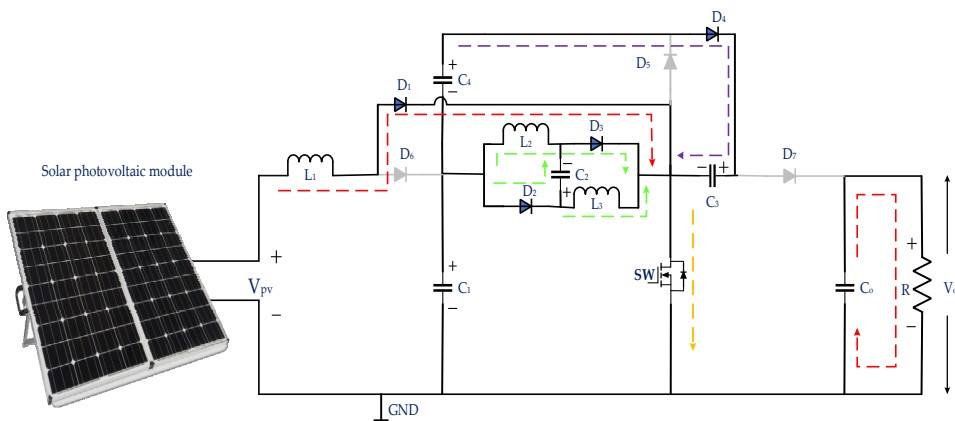

**Figure 7.** Operation of proposed HVGSU converter configuration during mode 1.

### 3.2. Operation of Proposed HVGSU Converter Configuration (Mode 2: SW Turn Off)

Figure 8 depicts the working principle during the second mode operation. At this time, the voltage will add up, and the gain will be transferred to the output via diodes $D_6$ and $D_7$. The inductor $L_2$ will charge the capacitor $C_4$ via diode $D_5$. In this mode, inductors $L_1$, $L_2$, and $L_3$ are energy storage inductors, and diodes $D_5$, $D_6$, and $D_7$ are only forward-biased. During this mode of operation, all other diodes will be biased in reverse.

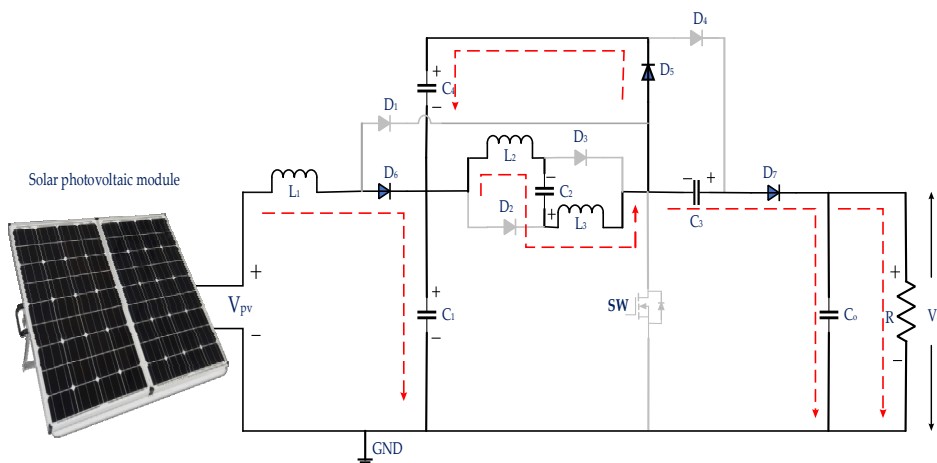

**Figure 8.** Operation of proposed HVGSU converter configuration during mode 2.

For the voltage and current relationships, there are some assumptions in the analysis:

1.  Existence of steady-state circumstances.
2.  The inductor current is uninterrupted (always positive).
3.  The switching period is T, and the switch is closed for DT and open for $(1 - D)$ T.
4.  The components are ideal.
5.  The capacitor is extremely large, and the output voltage is maintained at $V_o$.

The investigation continues by comparing the voltage and current of the inductor with the switch closed and open.

The analysis for the switch closed (mode 1).

$$V_{L1} = V_{pv} \tag{1}$$

$$V_{L3} = V_{L2} = V_{C1} = V_{C2} \tag{2}$$

$$V_{C4} - V_{C3} + V_{C2} = 0 \tag{3}$$

$$V_{C4} - V_{C3} = -V_{C1} \tag{4}$$

The analysis for the switch open (mode 2).

$$V_{pv} - V_{L1} - V_{C1} = 0 \tag{5}$$

$$V_{L1} = V_{pv} - V_{C1} \tag{6}$$

$$V_{C1} - V_{L2} + V_{C2} - V_{L3} + V_{C3} - V_o = 0 \tag{7}$$

$$2V_{L2} = 2V_{C1} + V_{C3} - V \tag{8}$$

$$V_{L2} = V_{C1} + \frac{V_{C3}}{2} + \frac{V_o}{2} \tag{9}$$

Average voltage across inductor $L_1$ in a steady state is equal to zero.

$$V_{pv}\,D + (V_{pv} - V_{C1})(1 - D) = 0 \tag{10}$$

$$V_{pv}\,D + V_{pv}(1 - D) - V_{L1}(1 - D) = 0 \tag{11}$$

$$V_{pv} - V_{C1}(1 - D) = 0 \tag{12}$$

$$V_{C1} = \frac{V_{pv}}{1 - D} \tag{13}$$

Average voltage across inductor $L_2$ in a steady state is equal to zero.

$$\frac{V_{pv}}{1 - D}D + \left(\frac{V_{pv}}{1 - D} + \frac{V_{C3}}{2} - \frac{V_o}{2}\right)(1 - D) = 0 \tag{14}$$

$$V_o = \frac{2V_{pv}}{(1 - D)^2} + \frac{2V_{pv}}{1 - D} + V_{C3} \tag{15}$$

Applying KVL,

$$V_{C1} + V_{C4} + V_{C3} - V_o = 0 \tag{16}$$

$$V_{C1} - V_o = -V_{C4} - V_{C3} \tag{17}$$

$$V_{C1} = V_{C3} - V_{C4} \tag{18}$$

$$-V_o = -V_{C3} - V_{C3} \tag{19}$$

$$V_{C3} = \frac{V_o}{2} \tag{20}$$

Putting the value of $V_{C3}$ from Equation (20) into Equation (15):

$$V_o = \frac{2V_{pv}D}{(1-D)^2} + \frac{2V_{pv}D}{1-D} + \frac{V_o}{2} \tag{21}$$

$$\frac{V_o}{2} = \frac{2V_{pv}D}{(1-D)^2} + \frac{2V_{pv}D}{1-D} \tag{22}$$

$$V_o = \frac{4V_{pv}D + (4V_{pv}D - 4V_{pv}D)}{(1-D)^2} \tag{23}$$

$$V_o = \frac{4V_{pv}}{(1-D)^2} \tag{24}$$

The voltage gain of the proposed HVGSU converter can be expressed as calculated in Equation (24).

## 4. Simulation and Experimental Results

### 4.1. Simulation Results

MATLAB Simulink software stimulates the proposed circuit to verify the converter's performance. The exact value for the circuit parameter was chosen as shown in Table 1. The simulation is run with input voltages of 20 V, 25 V, 30 V, 35 V, and 40 V at a switching frequency of 50 kHz. Every component has been chosen as an ideal component. The 150 W output power is used to determine the load resistance. The solar PV module specifications are shown in Table 2.

**Table 1.** Component parameters used in the MATLAB simulation for the proposed HVGSU converter.

| Parameter | Specification |
|---|---|
| PV module output voltage source | 20 V–40 V |
| Capacitors ($C_1$, $C_2$, $C_3$, $C_4$, $C_o$) | 150 µF, 240 µF, 250 µF, 250 µF, 100 µF |
| Inductor ($L_1$, $L_2$, $L_3$) | 300 µH, 600 µH, 600 µH |
| Load (R load) | 100 Ω |

**Table 2.** Solar PV module specifications.

| Parameter | Value |
|---|---|
| $V_{oc}$ | 50 V |
| $I_{sc}$ | 4.3 A |
| $V_{mp}$ | 40 V |
| $I_{mp}$ | 3.75 A |
| $P_{mp}$ | 150 W |

Figure 9 shows the simulated waveforms for the solar PV voltage ($V_{pv}$) and output voltage ($V_o$) for the proposed HVGSU converter. When the solar PV voltage ($V_{pv}$) is 20 V and the duty cycle is controlled to 0.54 in order to achieve a DC microgrid voltage $V_o = 380$ V (as shown in Figure 1), the duty cycle is smaller, which will reduce switching losses and make the converter more efficient.

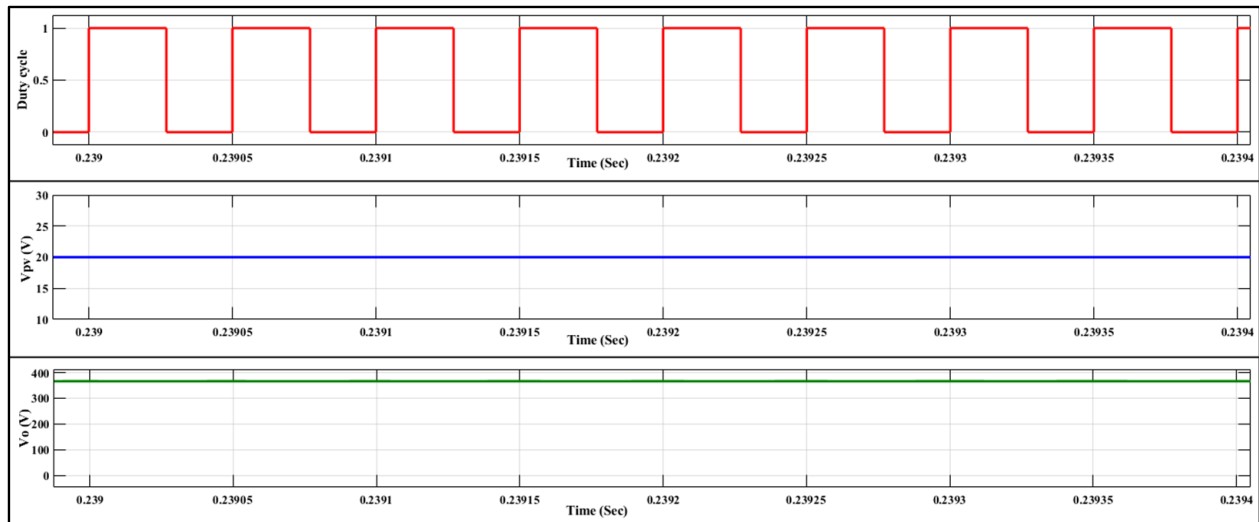

**Figure 9.** Simulated waveform for solar PV module output voltage ($V_{pv}$) and output voltage ($V_o$) of the proposed HVGSU converter at the 0.54 duty cycle.

Figure 10 shows the simulated waveforms for the solar PV voltage ($V_{pv}$) and output voltage ($V_o$) for the proposed HVGSU converter. When the solar PV voltage ($V_{pv}$) is 25 V and the duty cycle is controlled to 0.48 in order to achieve a DC microgrid voltage $V_o$ = 380 V (as shown in Figure 1), the duty cycle is smaller, which will reduce switching losses and make the converter more efficient.

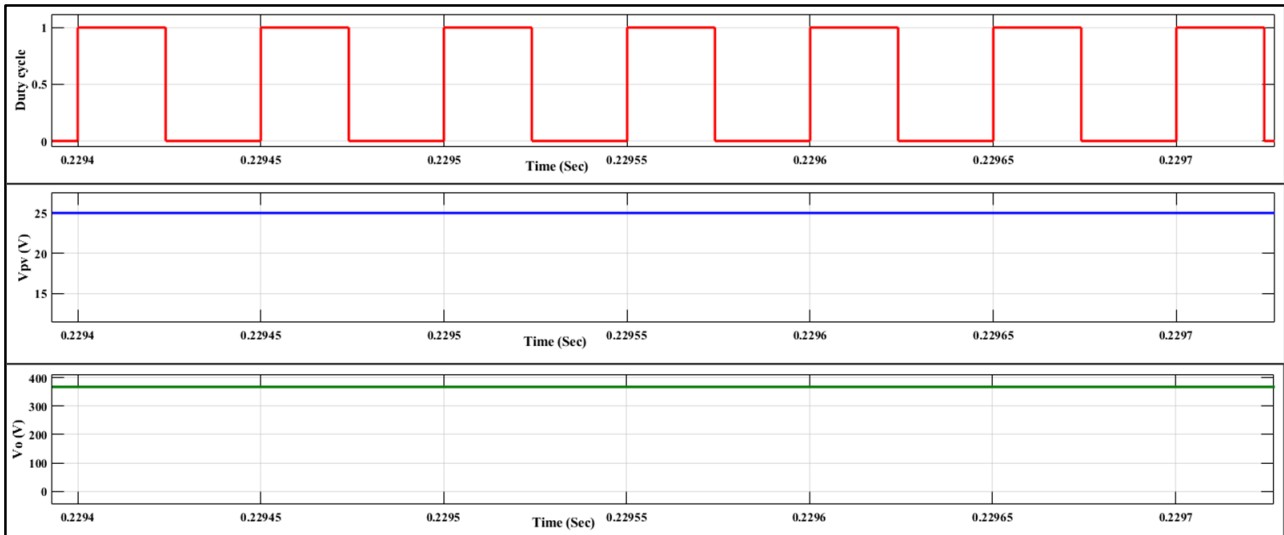

**Figure 10.** Simulated waveform for solar PV module output voltage ($V_{pv}$) and output voltage ($V_o$) of the proposed HVGSU converter at the 0.48 duty cycle.

Figure 11 shows the simulated waveforms for the solar PV voltage ($V_{pv}$) and output voltage ($V_o$) for the proposed HVGSU converter. When the solar PV voltage ($V_{pv}$) is 30 V and the duty cycle is controlled to 0.44 in order to achieve a DC microgrid voltage $V_o$ = 380 V (as shown in Figure 1), the duty cycle is smaller, which will reduce switching losses and make the converter more efficient.

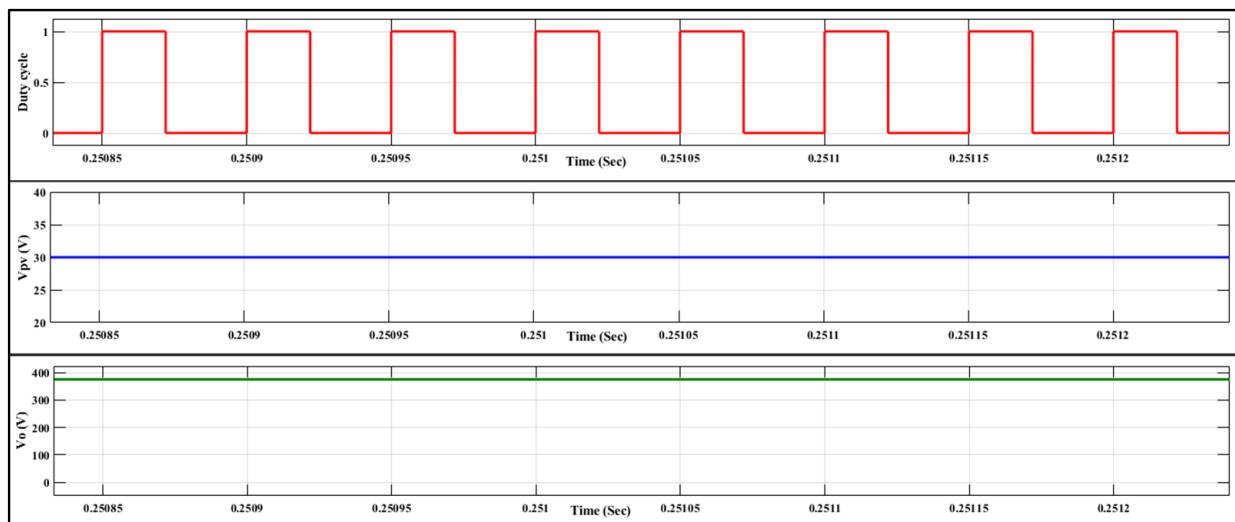

**Figure 11.** Simulated waveform for solar PV module output voltage ($V_{pv}$) and output voltage ($V_o$) of the proposed HVGSU converter at the 0.44 duty cycle.

Figure 12 shows the simulated waveforms for the solar PV voltage ($V_{pv}$) and output voltage ($V_o$) for the proposed HVGSU converter. When the solar PV voltage ($V_{pv}$) is 35 V and the duty cycle is controlled to 0.39 in order to achieve a DC microgrid voltage $V_o$ = 380 V (as shown in Figure 1), the duty cycle is smaller, which will reduce switching losses and make the converter more efficient.

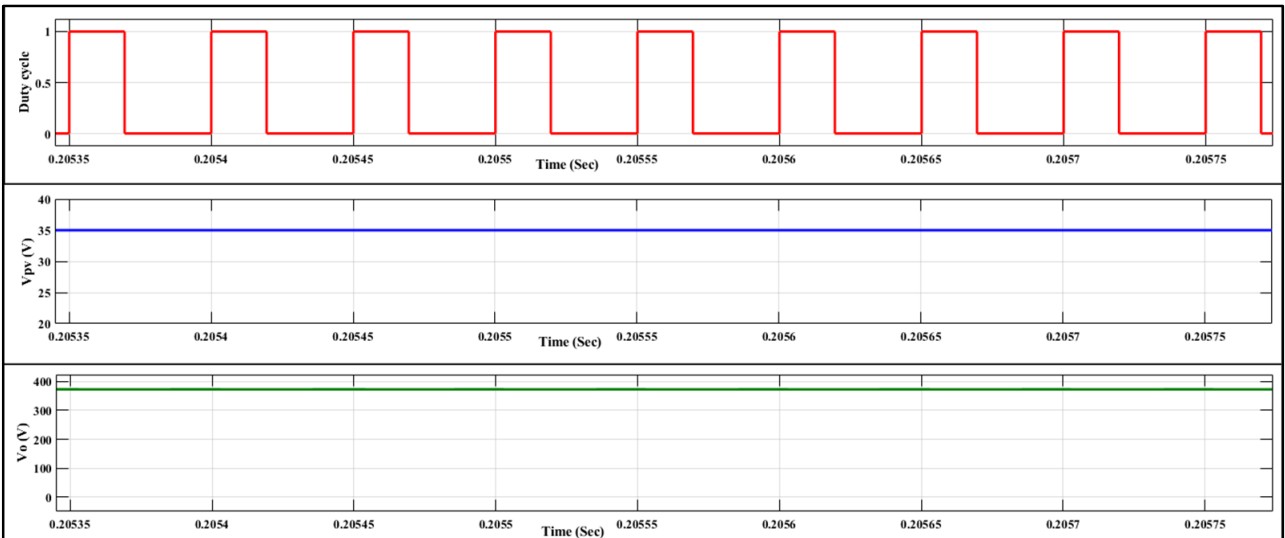

**Figure 12.** Simulated waveform for solar PV module output voltage ($V_{pv}$) and output voltage ($V_o$) of the proposed HVGSU converter at the 0.39 duty cycle.

Figure 13 shows the simulated waveforms for the solar PV voltage ($V_{pv}$) and output voltage ($V_o$) for the proposed HVGSU converter. When the solar PV voltage ($V_{pv}$) is 40 V and the duty cycle is controlled to 0.35 in order to achieve a DC microgrid voltage $V_o$ = 380 V (as shown in Figure 1), the duty cycle is smaller, which will reduce switching losses and make the converter more efficient and catch the MPP.

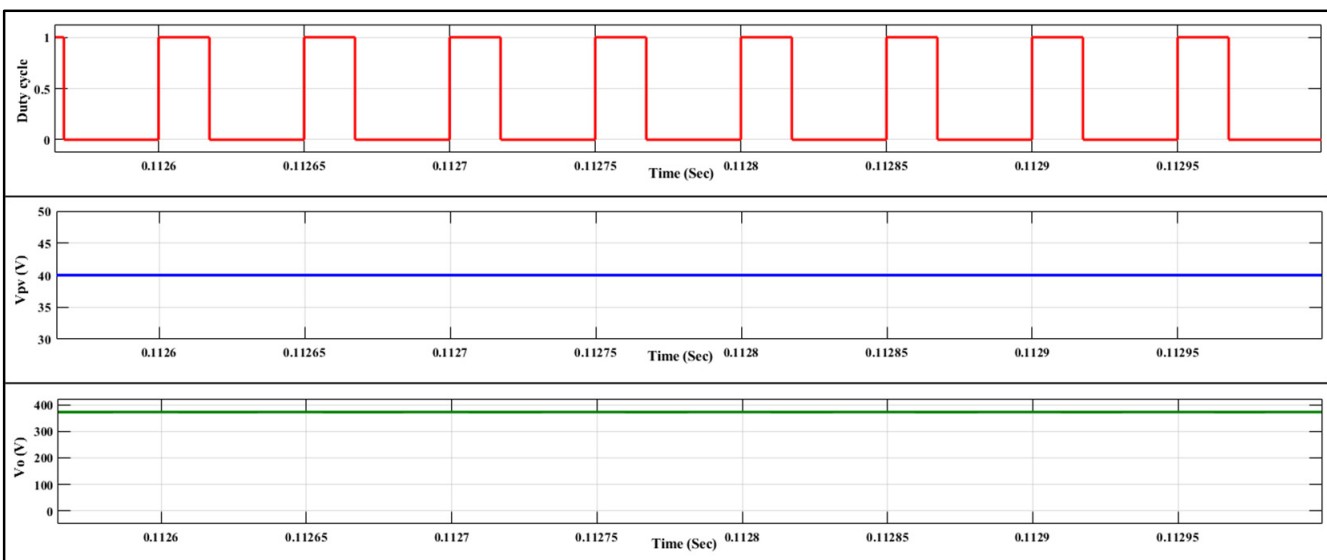

**Figure 13.** Simulated waveform for solar PV module output voltage ($V_{pv}$) and output voltage ($V_o$) of the proposed HVGSU converter at the 0.35 duty cycle.

From Figures 9–13, the values of $V_{pv}$ are 20 V, 25 V, 30 V, 35 V, and 40 V, respectively. The duty cycle of the proposed novel high-voltage gain step-up (HVGSU) converter is adjusted through Equation (24). The output voltage produced by the HVGSU converter is stable at 380 V.

This study proposes a novel 150 W HVGSU DC–DC converter to fulfill the maximum power requirements of solar PV modules with a maximum output of 150 W (as shown in Table 2). The input voltage for this simulation is set to 20 V, 25 V, 30 V, 35 V, and 40 V, because the output efficiency of the solar PV module will vary substantially, depending on the climate in the actual work environment. This research uses the proposed HVGSU converter and adjusts the duty cycle according to Equation (24) so that the output voltage can operate at 380 V.

*4.2. Hardware Results*

In this section, the HVGSU converter based on the proposed model is developed, presented, and investigated at the Power & Energy Research Lab, National Taiwan University of Science and Technology (Taiwan Tech). Figure 14 shows the structural block diagram of the proposed HVGSU converter embedded with the hill climbing control strategy. The output voltage is regulated at 380 V, and the solar PV voltage ($V_{pv}$) ranges between 20 V and 40 V. The test setup is depicted in Figure 15. Table 3 lists the specifications of the hardware components in tabular format, and Table 2 gives the specifications of the solar PV simulator; in this case, the solar irradiance G = 1000 $W^2$/m and temperature T is 25 °C. The duty ratio was computed for the specific output voltage rating ($V_o$ Ideal = 380 V as Equation (24)), and the PWM signal was generated by the microcontroller unit (MCU) F28004xC2000. The DC electronic load selects a continuous current mode for the load. The experimental waveform is obtained from the digital oscilloscope and is present later step by step.

Figure 16 shows the measured waveforms for the solar PV simulator ($V_{pv}$) and output voltage ($V_o$) for the proposed HVGSU converter, with an input voltage of 20 V and a duty cycle of 0.54, in order to generate a DC microgrid voltage of 380 V (as shown in Figure 1). The experimental results show that the duty cycle is small, which reduces the conduction losses and enhances the converter efficiency.

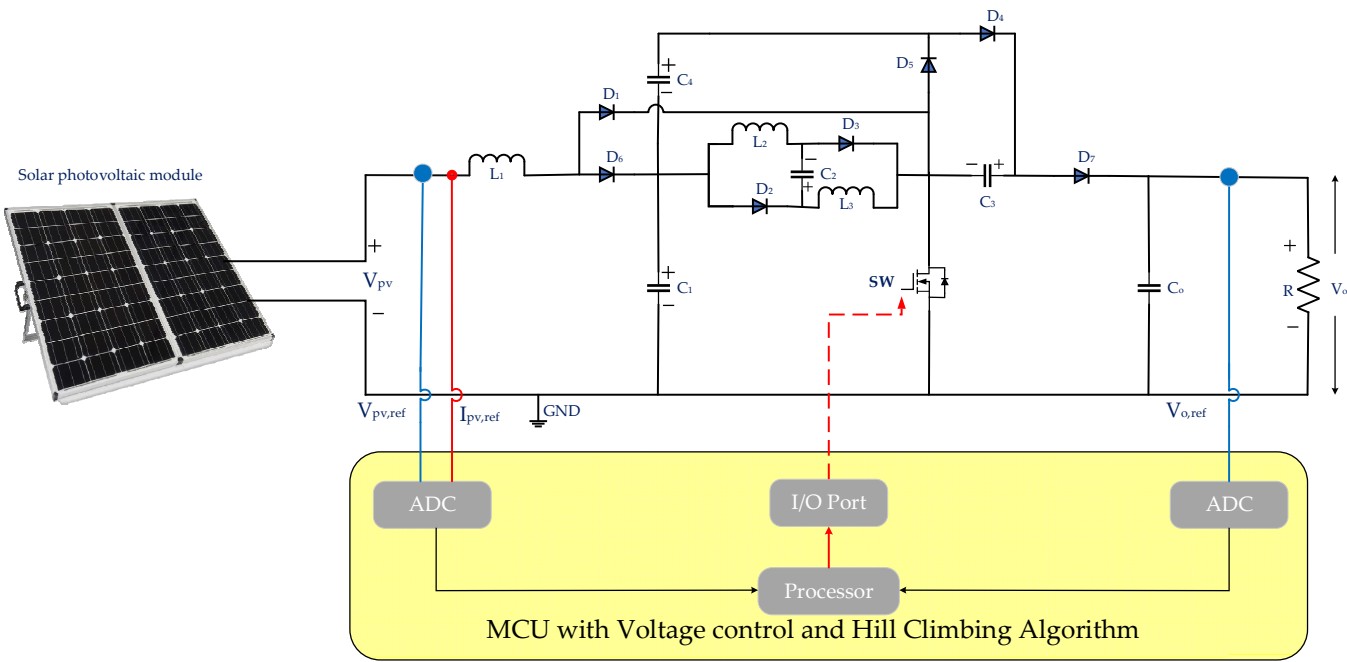

**Figure 14.** Structural block diagram of the proposed HVGSU converter embeds with the hill climbing control strategy.

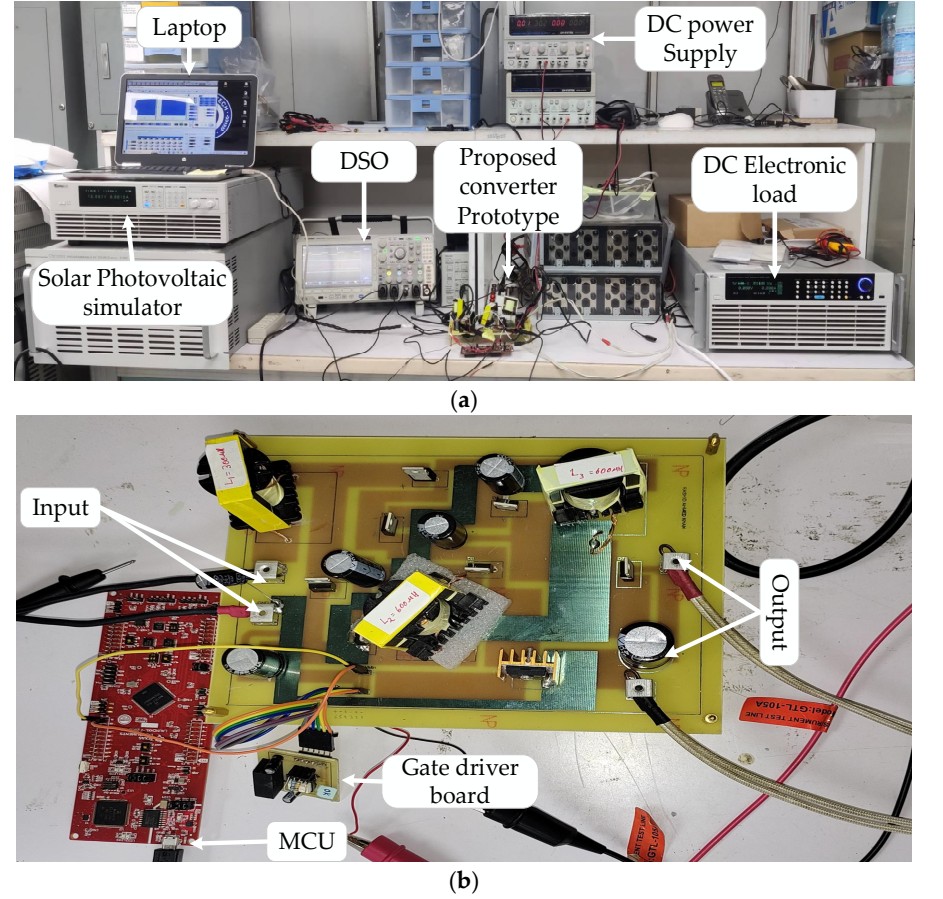

**Figure 15.** (**a**) Experimental testbed of the proposed HGVSU converter in a power and energy laboratory, and (**b**) hardware prototype of the proposed HGVSU converter.

**Table 3.** The component specifications used in the hardware prototype of the proposed converter.

| Components | Specification |
|---|---|
| Capacitors ($C_1$, $C_2$, $C_3$, $C_4$, $C_o$) | 150 μF, 240 μF, 250 μF, 250 μF, 100 μF |
| Inductor ($L_1$, $L_2$, $L_3$) | 300 μH, 600 μH, and 600 μH |
| Microcontroller (Texas instruments) | F28004xC2000 |
| Diode | 52N50C3 HFV138 |
| MOSFET | PFCD86G |
| Gate Driver IC | TLP250H |
| Solar PV Simulator | Chroma, 62100H-600S |
| Load Resistance (DC Electronic load) | Chroma, 63204A-600-280 |

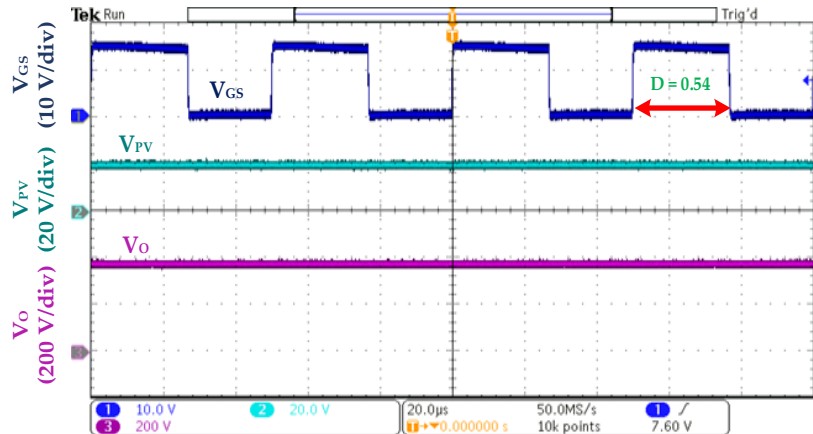

**Figure 16.** Experimental waveform for gate source voltage ($V_{GS}$), solar photovoltaic simulator output voltage ($V_{PV}$), and output voltage ($V_o$) for the proposed HVGSU converter at the 0.54 duty cycle (Hor: 20 μs/div).

Figure 17 depicts the measured waveforms for the solar PV simulator ($V_{pv}$) and output voltage ($V_o$) for the proposed HVGSU converter, with an input voltage of 25 V and a duty cycle of 0.48, in order to generate a DC microgrid voltage of 380 V (as shown in Figure 1). In this test condition, the duty cycle is also tiny to reduce the conduction losses and improve the converter efficiency.

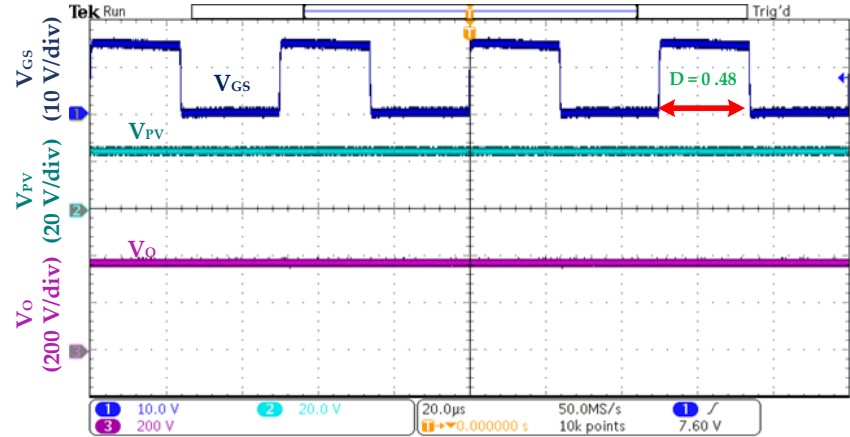

**Figure 17.** Experimental waveform for the gate source voltage ($V_{GS}$), solar photovoltaic simulator output voltage ($V_{PV}$), and output voltage ($V_o$) for the proposed HVGSU converter at the 0.48 duty cycle (Hor: 20 μs/div).

Figure 18 displays the measured waveforms for the solar PV simulator ($V_{pv}$) and output voltage ($V_o$) for the proposed HVGSU converter, with an input voltage of 30 V and a duty cycle of 0.44, in order to generate a DC microgrid voltage of 380 V (as shown in Figure 1). When the input voltage is increased to 30 V, the duty cycle is still small, which reduces the conduction losses and enhances the converter efficiency.

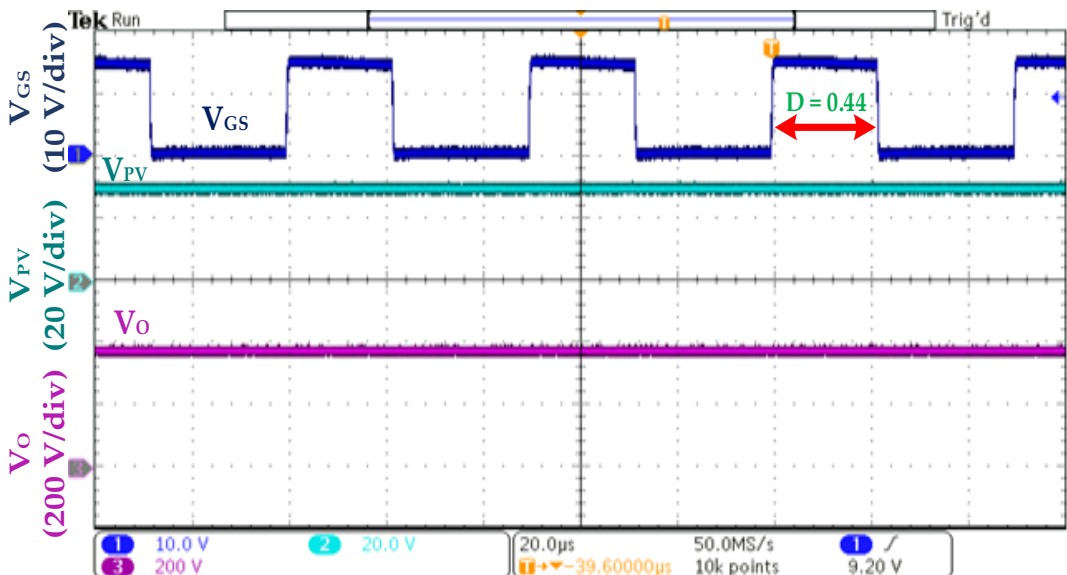

**Figure 18.** Experimental waveform for the gate source voltage ($V_{GS}$), solar photovoltaic simulator output voltage ($V_{PV}$), and output voltage ($V_o$) for the proposed HVGSU converter at the 0.44 duty cycle (Hor: 20 µs/div).

Figure 19 shows the measured waveforms for the solar PV simulator ($V_{pv}$) and output voltage ($V_o$) for the proposed HVGSU converter, with an input voltage of 35 V and a duty cycle of 0.39, in order to generate a DC microgrid voltage of 380 V (as shown in Figure 1). Under these conditions, the duty cycle is small, reducing the conduction losses and raising the converter efficiency.

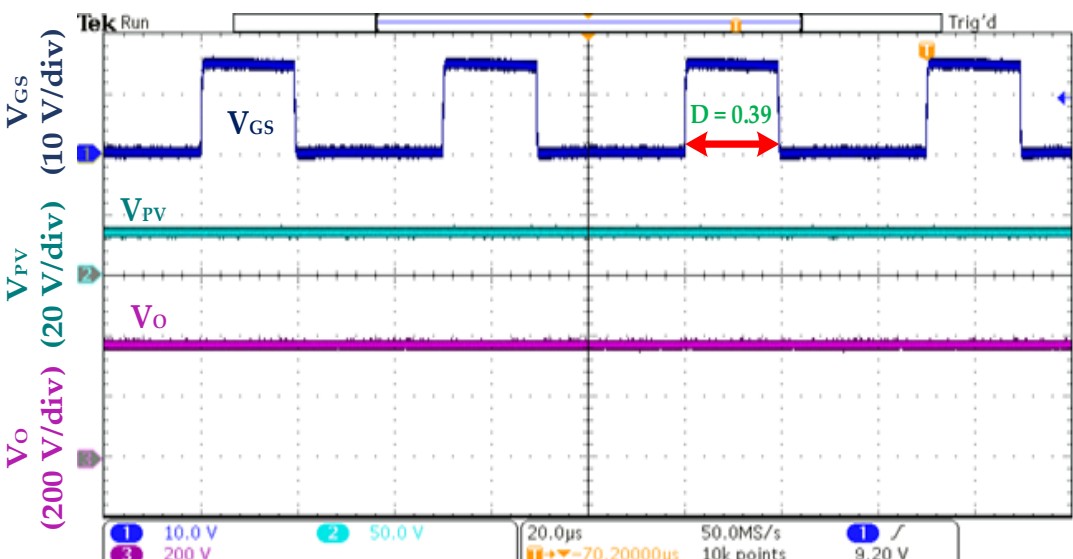

**Figure 19.** Experimental waveform for the gate source voltage ($V_{GS}$), solar photovoltaic simulator output voltage ($V_{PV}$), and output voltage ($V_o$) for the proposed HVGSU converter at the 0.39 duty cycle (Hor: 20 µs/div).

Figures 16–19 show the real test of the solar photovoltaic module connected to the proposed HVGSU converter. The test conditions include $V_{pv}$ of 20 V, 25 V, 30 V, and 35 V; the $V_o$ result is 380 V in all cases, as in Equation (24).

Figure 20 shows the real test of the solar photovoltaic module connected to the proposed HVGSU converter and built with the hill climbing (HC) algorithm. When the HC algorithm is used, the MPP duty cycle is 0.35 and the $V_{pv}$ is 40 V, which is the same as the $V_{mp}$ in Table 2. The output voltage $V_o$ is 380 V, based on Equation (24), to generate a DC microgrid voltage of 380 V (as shown in Figure 1). Even if the input voltage is up to 40 V, the duty cycle keeps small, which reduces the conduction losses and improves the converter efficiency. In this condition, the hill climbing algorithm is implemented.

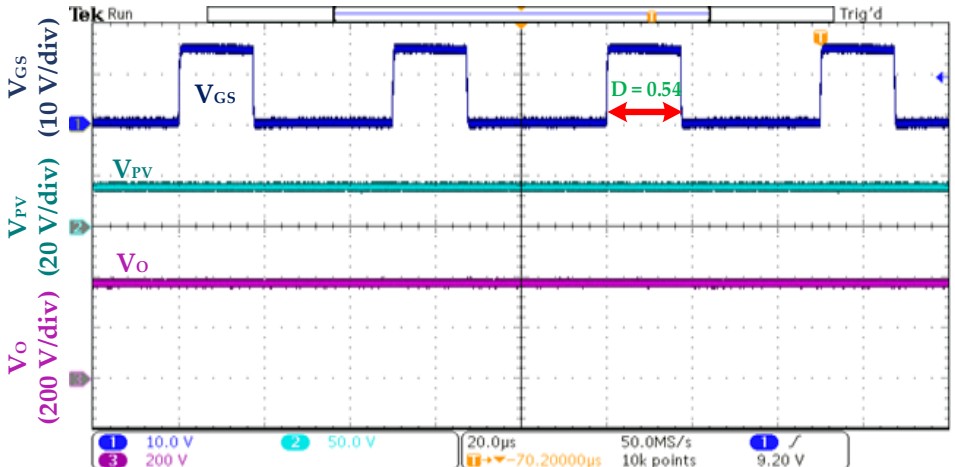

**Figure 20.** Experimental waveform for the gate source voltage ($V_{GS}$), solar photovoltaic simulator output voltage ($V_{PV}$), and output voltage ($V_o$) for the proposed HVGSU converter at the 0.35 duty cycle (Hor: 20 μs/div).

This study aims to demonstrate the high-gain ratio of the proposed novel high-voltage gain step-up (HVGSU) DC–DC converter. Therefore, the simulation and experimental results show the $V_{GS}$, $V_{PV}$, and $V_o$. The output voltage of this HVGSU converter is a high voltage level, which conforms to Equation (24). Further, the test data and efficiency are presented in Table 4.

**Table 4.** Experimental results.

| Condition | $V_{PV}$ (V) | $V_o$ (V) | Duty Cycle | Efficiency |
|:---:|:---:|:---:|:---:|:---:|
| 1 | 20 | 380 | 0.54 | 90% |
| 2 | 25 | 380 | 0.48 | 91% |
| 3 | 30 | 380 | 0.44 | 93% |
| 4 | 35 | 380 | 0.39 | 94% |
| 5 | 40 | 380 | 0.35 | 96% |

The outcomes of the experiment are discussed and illustrated in Table 4. The duty ratio is determined as 0.54, 0.48, 0.44, 0.39, and 0.35 using the ideal voltage gain equation (i.e., Equation (24)) to extract the output voltage ($V_o$) around 380 V from the chosen input voltages ($V_{pv}$) of 20 V, 25 V, 30 V, 35 V, and 40 V. We can see that the efficiency varied from 90 to 96% at different conditions. The tolerance for the output voltage $V_o$ is ±5% of the 380 V DC microgrid voltage (as shown in Figure 1). Since the actual converter's voltage gain adheres to a non-ideal voltage gain, the corresponding voltage gain deviates slightly from the ideal gain.

### 4.3. Comparison of the Ideal Voltage Gain of the High-Gain DC–DC Boost Converter

Figure 21 briefly compares the ideal voltage gains of a recently proposed non-isolated based architecture. Among the potential topologies, the group of Mohammad et al. [33] suggested a VMC-based topology with a (3D) timed quadratic-based boost voltage gain. The configuration described in Section 1 not only improves the voltage gain but also reduces the switch's voltage stress (i.e., switch's voltage stress is 2 $V_o$/(3D) [33] and proposed topology $V_o$/2. It can also be concluded from Figure 21 that the voltage gain of the proposed converter is much better that the other recently proposed HVGSU converter. Table 5 compares the seven distinct topologies and concludes that the proposed circuit offers superior performance and cost advantages.

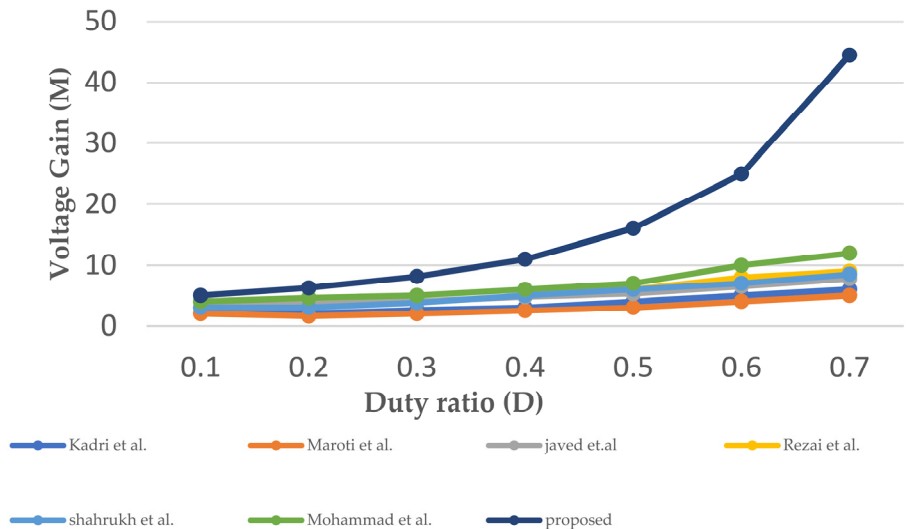

**Figure 21.** Voltage gain vs. duty ratio plot of the recently proposed topology [14,22,34,36,40,45].

**Table 5.** Comparison of different types of converters in terms of cost, voltage stress, and efficiency.

| Type | Voltage Stress | Efficiency | Cost |
|---|---|---|---|
| [14] | High | Low | Medium |
| [22] | High | Low | Medium |
| [33] | Medium | Medium | Medium |
| [35] | High | Low | Medium |
| [39] | High | Low | Medium |
| [44] | Medium | Medium | Medium |
| Proposed converter | Low | High | Medium |

Table 5 presents a comparison of various types of existing proposed converters in terms of various parameters, such as cost, efficiency, and voltage stress. It was encountered that the proposed converter has lower voltage stress when compared to [14,22,33,35,39,44], and its efficiency is also greater than [14,22,34,36,40,45].

## 5. Conclusions and Future Works

Within the scope of this study, an HVGSU converter with a minimum number of switches, extendable gain, and with minimum normalized voltage stress has been proposed, developed, and investigated. The experimental results and the correct designing of all components, including the PCB, using Altium PCB Designer have allowed for the verification of the exemplary performances associated with the proposed topology. The voltage stress across the switch is directly connected to the series integrated capacitor C3

(i.e., refer to Figure 4), which significantly reduces the voltage stress across the switch up to 50% of the output voltage. When all the losses associated with the various components are considered, the suggested topology has demonstrated a peak efficiency of around 90–96% for $V_{in}$ of 20 V to 40 V and $V_{out}$ of 380 V. The circuit is also tested with an input voltage of 20 V, 25 V, 30 V, 35 V, and 40 V and can give an output voltage between 380 V and 400 V. The converter simultaneously illustrated the benefit of a high-voltage gain, low-input ripple current, and low-voltage stress. Therefore, the proposed topology was well suited for the DC nanogrid to obtain high step-up gain and efficiency. The simulation and experiment results were well aligned with the theoretical studies, proving the proposed HVGSU converter has credibility as a suitable high-voltage gain power electronic converter for photovoltaic applications. Therefore, the proposed family of converters is good for applications such as renewable energy, microgrids, and uninterruptible power supplies (as shown in Figure 1). The proposed converter has demonstrated its potential for high-boost applications.

The proposed converter can step down voltages by adding open-zero states to the controller gate pulses. This is something that almost none of the high-gain DC–DC converters have. This is a useful feature, because it makes the converter useful for a wider range of tasks. The research in this paper was mostly about one type of power electronic converter. In future works, it might be possible to study another renewable energy system, such as wind energy or fuel cells, that uses the proposed converter and the new controlled strategy with the whole system.

**Author Contributions:** Conceptualization, R.A.K. and H.-D.L.; Formal analysis, R.A.K.; Investigation, R.A.K. and C.-H.L.; Software, R.A.K. and H.-D.L.; Methodology, R.A.K., A.S., C.-H.L. and H.-D.L.; Data curation, R.A.K., H.-D.L. and A.S.; Visualization, R.A.K., A.S. and H.-D.L.; Funding acquisition, C.-H.L.; Supervision, A.S., H.-D.L., S.-D.L., S.-J.Y. and C.-H.L.; Writing—original draft, R.A.K.; and Writing—review and editing, R.A.K., A.S. and H.-D.L. All authors have read and agreed to the published version of the manuscript.

**Funding:** This research was funded by the National Science and Technology Council, Taiwan, R.O.C., grant number NSTC 111-2221-E-011-071, NSTC 111-3116-F-011-005, and NSTC 111-2622-8-005-003-TE1. The authors also sincerely appreciate the considerable support from the National Taiwan Normal University Subsidy Policy for International Collaboration and Research Projects.

**Institutional Review Board Statement:** Not applicable.

**Informed Consent Statement:** Not applicable.

**Data Availability Statement:** Not applicable.

**Acknowledgments:** The authors acknowledge the National Science and Technology Council, Taiwan, R.O.C., grant number NSTC 111-2221-E-011-071, NSTC 111-3116-F-011-005, and NSTC 111-2622-8-005-003-TE1. The authors also sincerely appreciate the partial financial support from the Taiwan Building Technology Center from The Featured Areas Research Center Program within the framework of the Higher Education Sprout Project by the Ministry of Education in Taiwan.

**Conflicts of Interest:** The authors declare no conflict of interest.

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
