# Peer review of "A Novel High-Voltage Gain Step-Up DC–DC Converter with Maximum Power Point Tracker for Solar Photovoltaic Systems"

_processes, doi:10.3390/pr11041087_

Round 1

Reviewer 1 Report

Title: A Novel High Voltage Gain Step-Up DC-DC Converter with Maximum Power Point Tracker for Solar Photovoltaic Systems

1)    Why is the clamp circuit used in the proposed high voltage gain step-up DC-DC converter? Explain more about this.

2)    Why hill climbing algorithm is used in the maximum power point tracker? Also, the hill climbing algorithm seems to be not comprehensively investigated in the literature.

3)    Why in the simulation, the input voltage is set to 20 V, 25 V, 30 V, 35 V and 40 V, and the duty cycle is continuously reduced to obtain the target output voltage of 380V? How to set this? Besides, why is the output power set to 150w?

4)    The structural block diagram of the hardware circuit in Figure 13 is not given.

5)    Figures 8-12 seem to be all the same, in which Vo and Vpv are constant, how about their changes?

6)    Why is there only a comparison of the performance index of voltage gain at different duty cycles compared with previous research in Figure 19, without a detailed comparison of its efficiency and so on?

Author Response

1) Many thanks for Reviewer’s valuable suggestions. We have added some explanations of this revised manuscript below.

The passive clamp can absorb the energy loss due to the inductor's leaking inductance to prevent an excessive voltage spike at the switch. The passive clamp and voltage multiplier circuit arrangement boost the voltage gain. The advantages of this converter include its high efficiency, low switching loss, high voltage gain at a low-duty cycle, low turn ratio for the inductor, low voltage stress on the switch, and low voltage stress on the diodes. The details have been added in Lines 200-205, Page 5.

2) Many thanks for Reviewer’s valuable suggestions. Our explanations of this problem is as follows.

  1. The HC algorithm’s principle and description have been introduced in section 2 and Fig. 3.
  2. This algorithm is effective for environments with uniform sunshine but not for environments with shade. This research employs the novel high voltage gain step-up (HVGSU) DC-DC converter, which has a high boost effect, to enable the parallel operation of solar PV modules. Even if the shading issue arises, the Efficiency will not be affected. The HVGSU converter proposed in this experiment in conjunction with the HC algorithm, can operate in a partial shade environment and mitigate the algorithm's inadequacies.

The details have been added in Lines 240-256, Pages 6 and 7.

3) Many thanks to the Reviewer’s reminder. We have strengthened the related instructions of this revised manuscript below.

  1. This study proposes a novel 150W HVGSU DC-DC converter to fulfill the maximum power requirements of solar PV modules with a maximum output of 150W. (as Table 2).
  2. The input voltage for this simulation is set to 20V, 25V, 30V, 35V, and 40V because the output efficiency of the solar PV module will vary substantially depending on the climate in the actual work environment. This research uses the proposed HVGSU converter and adjusts the duty cycle according to equation (24) so that the output voltage can operate at 380V.

The details have been added in Lines 402-408, Page 14.

4) Many thanks for Reviewer’s valuable suggestions. We have added the structural block diagram as shown in Fig. 13, Page 15.

5) Many thanks to the Reviewer’s reminder. We have strengthened the related explanation of this revised manuscript as follows.

From Figure 8 to Figure 12, the Value of Vpv are 20V, 25 V, 30 V, 35 V, and 40V, respectively. The duty cycle of the proposed novel high voltage gain step-up (HVGSU) converter is adjusted through equation (24). The output voltage produced by the HVGSU converter is stable at 380 V. The details have been added in Lines 398-401, Page 14.

6) Many thanks for Reviewer’s valuable suggestions. We have added Table 5 and explained the efficiency and cost with other studies as follows.

Table 5 presents a comparison of various types of existing proposed converters in terms of various parameters such as cost, efficiency, and voltage stress. It was encountered that the proposed converter has lower voltage stress when compared to [14], [22], [34], [36], [40], and [45], and its efficiency is also greater than [14], [22], [34], [36], [40], and [45]. The details have been added in Lines 520-526, Page 20.

Reviewer 2 Report

1. Abstract, Introduction and Conclusion are not well organized and should be shorten.

2. The hill climbing MPPT method is not novel.

3. The simulation and experimental results are not edequate, only VGS, VPV and Vo under different duty cycles are presented.

4. The voltage rage is 12V-60V in line 105, which is inconsistent with Figure 1 (20V-40V).

5. Grammar errors are present, such as "The proposed a novel high voltage gain step-up (HVGSU) DC-DC converter with ultrahigh voltage gain that considers pertinent features." in line 253-254.

Author Response

1) Many thanks for Reviewer’s valuable suggestions. The Abstract, Introduction, and Conclusion have been rewritten to make them organized and shortened. The details have been added in Lines 16-32, 107-186, 528-554, Pages 1, 3-5, and 20.

2) Many thanks to the Reviewer’s reminder. In particular, this study uses the HC algorithm, combined with the proposed novel high voltage gain step-up (HVGSU) DC-DC converter. To improve the shortcomings of the HC algorithm, we have added some explanations of this revised manuscript below.

This algorithm is suitable for a uniform sunlight environment, but not for a shading environment. This research uses the proposed novel high voltage gain step-up (HVGSU) DC-DC converter, which has a high boost effect, so solar PV modules can be used in parallel. Even if the shading problem occurs, it will not affect the system’s Efficiency. The HVGSU converter proposed in this experiment combined with the HC algorithm can work in a sheltered environment and improve the shortcomings of the HC algorithm. The details have been added in Lines 248-254, Pages 6-7. 

3) Many thanks to the Reviewer’s reminder. We have strengthened the related explanation of this revised manuscript as follows.

This study aims to demonstrate the high gain ratio of the proposed novel high voltage gain step-up (HVGSU) DC-DC converter. Therefore, the simulation and experimental results show VGS, VPV, and Vo. The output voltage of this HVGSU converter is a high voltage level, which conforms to equation (24). Further, The test data and efficiency are presented in Table 4. The details have been added in Lines 491-495, Pages 18-19.

4) Many thanks for Reviewer’s valuable suggestions. We have modified the voltage range of this revised manuscript. The details have been added in Line 99, Page 3.

5) Many thanks for Reviewer’s valuable suggestions. We have modified the paragraph of this revised manuscript. The details have been added in Lines 258-261, Page 7.

Round 2

Reviewer 2 Report

1. The sentence "and the duty cycle is smaller, which will reduce switching losses and make the converter more efficient and catch the MPP" repeats in each paragraph after the simulated waveform, and the sentence "However, the high duty cycle is small, which reduces switching losses and enhances converter efficiency" repeats in each paragraph after the experimental result.

These do not sound technical, please improve the writing. In additon, the switching loss can be reduced by decrease the switching frequency instead of the duty cycle, and what does "high duty cycle is small" mean?

2. In Fig. 5, VC3max and VC3min should be VC4max and VC4min in the waveform of VC4, respectively.

Author Response

1) Many thanks for Reviewer’s valuable suggestions. We have explained and corrected the above sentence as follows:

So as the output voltage should increase with the duty cycle, the power MOSFET would conduct current for a longer time as the duty cycle increases. So that the power MOSFET losses are increasing, and hence the efficiency decreases. We also edited the manuscript and the details have been added in Lines 449-553, 458-462, 467-471, 476-480, 485-491, Pages 16-18.

2) Many thanks to the Reviewer’s reminder. We have updated Fig. 5 of this revised manuscript.